# Brown Adipose Tissue and Its Role in Insulin and Glucose Homeostasis

**DOI:** 10.3390/ijms22041530

**Published:** 2021-02-03

**Authors:** Katarzyna Maliszewska, Adam Kretowski

**Affiliations:** Department of Endocrinology, Diabetology and Internal Medicine, Medical University of Bialystok, 15-089 Bialystok, Poland; adamkretowski@wp.pl

**Keywords:** brown adipose tissue, obesity, diabetes type 2, insulin resistance, metabolism

## Abstract

The increased worldwide prevalence of obesity, insulin resistance, and their related metabolic complications have prompted the scientific world to search for new possibilities to combat obesity. Brown adipose tissue (BAT), due to its unique protein uncoupling protein 1 (UPC1) in the inner membrane of the mitochondria, has been acknowledged as a promising approach to increase energy expenditure. Activated brown adipocytes dissipate energy, resulting in heat production. In other words, BAT burns fat and increases the metabolic rate, promoting a negative energy balance. Moreover, BAT alleviates metabolic complications like dyslipidemia, impaired insulin secretion, and insulin resistance in type 2 diabetes. The aim of this review is to explore the role of BAT in total energy expenditure, as well as lipid and glucose homeostasis, and to discuss new possible activators of brown adipose tissue in humans to treat obesity and metabolic disorders.

## 1. Introduction

Brown adipose tissue (BAT) was considered, for several years, to be present only in newborns and small mammals to generate heat through non-shivering thermogenesis as protection against hypothermia. However, the abundant amount of active BAT in children declines rapidly after puberty. The exact amount (volume) of active BAT in adult humans remains highly variable, but the prevalence of brown adipose tissue in adults was estimated at 6.97% based on recently published results from a systematic review and meta-analysis [1]. The first clinical observations of BAT came from oncological patients in whom imaging scans, using positron emission tomography combined with computed tomography (PET/CT) or magnetic resonance PET/MR, revealed cervical adipose tissue characterized by high metabolic activity [2]. In 2009, functional brown fat in adult humans was confirmed after dedicated cold exposure research [3,4,5].

The increased worldwide prevalence of obesity has prompted the scientific world to search for new possibilities to deal with weight gain [6]. Obesity is a major health risk factor and strongly associated with the development of insulin resistance, which is a key player in the pathogenesis of metabolic complications, type 2 diabetes, and cardiovascular diseases [7]. An increased obesity rate is associated with a decrease in life expectancy and also represents a large economic burden [8].

BAT is a type of tissue designed for maintaining body temperature higher than ambient temperatures through heat production, primarily via non-shivering thermogenesis. This process is mediated by the expression of uncoupling protein 1 (UCP1) within the inner membrane of the abundant mitochondria [9]. Despite high mitochondria content and high cellular respiration rates, brown adipocytes have a remarkably low capacity for adenosine triphosphate (ATP) synthesis [10]. Brown adipocytes (in contrast to most human cells), through UCP1 expression and low ATP synthase activity, diminish the proton gradient by uncoupling cellular respiration and decrease mitochondrial ATP synthesis to stimulate heat production.

Due to its unique UPC1, brown adipose tissue has been acknowledged as a promising approach to increase energy expenditure [11]. In other words, BAT burns fat and increases the metabolic rate, promoting a negative energy balance [12]. Moreover, BAT alleviates metabolic complications like dyslipidemia, impaired insulin secretion, and insulin resistance in type 2 diabetes [13].

The protective role of BAT, in terms of its metabolic consequences, prompted the molecular exploration of brown adipocyte differentiation. The most relevant molecular factors involved in brown and white adipose tissue formation are peroxisome proliferator-activated receptors (PPARs) [14]. PPARγ has a crucial role in tissue development and functions by inducing UCP1 expression during adipogenesis [15]. Moreover, PPARγ agonists can be used to induce the browning of white adipose tissue [16], while PPARα activation promotes beige adipogenesis via Peroxisome proliferator-activated receptor gamma coactivator 1-alpha (PGC1α), which is a key regulator of mitochondrial biogenesis, adaptive thermogenesis, and oxidative metabolism. In the molecular pathways involved in white adipose tissue (WAT) browning, the key factor is PR domain-containing protein 16 (PRDM16), which controls the switch between skeletal myoblasts and brown adipocytes and stimulates adipogenesis by directly binding to PPARγ [17]. Recently, it was shown that brown and beige adipocytes release growth and differentiation factor 15 (GDF15) in response to thermogenic activity. GDF15 may mediate the downregulation of local inflammatory pathways [18]. Moreover, in adipose tissue biology, certain microRNAs play the important role of regulating BAT and WAT functions and differentiation. Such microRNAs regulate white, brown, and beige adipogenesis by targeting key transcription factors (e.g., PRDM16, PPARγ, CCAAT-enhancer binding protein C/EBPB, and PGC1α) [19].

The aim of this review is to explore the role of BAT in whole-body energy expenditure and lipid and glucose homeostasis and to discuss new possible activators of brown adipose tissue in humans to treat obesity and metabolic disorders.

## 2. Morphology of Brown Adipocytes

Brown adipose tissue (BAT) and white adipose tissue (WAT) contribute to total adipose tissue in humans. Despite having similar structural components (adipocytes), the functions of both fat tissues are different. White adipose tissue stores energy, while brown adipose tissue generates body heat. Brown adipose tissue consists of brown adipocytes (which are smaller than white adipocytes) and lipids that are located in multiple small droplets, giving brown adipocytes a multilocular histology [20] with a central nucleus and an abundant number of mitochondria. Dense vascularity and innervation by the sympathetic nervous system are characteristic features of brown adipocytes [21]. White adipocytes have a unilocular morphology, and triacylglycerols are stored in one large droplet inside the cell. Deposits of WAT are localized mainly beneath the skin (subcutaneous adipose tissue (SAT)) and around internal organs (visceral adipose tissue (VAT)). However, a small amount of WAT is found in the perivascular and epicardial regions, the mediastinal retro-optical space, and bone marrow [22]. The main function of white adipocytes is to act as a reserve for lipids, which can be oxidized to produce energy and protect the human body from excess glucose by storing triglycerides [23]. Among the white adipocytes are scattered beige (brite) adipocytes, which can be converted to brown adipocytes. The browning of WAT is induced by cold exposure or genetic modification. Brite adipocytes share morphological and molecular features and functions with typical BAT.

The traditional understanding of white fat tissue as passive energy storage is no longer justifiable. In recent years, adipose tissue has been recognized as a complex, highly active metabolic and endocrine organ [24], with special emphasis on visceral fat as being more prone to induce insulin resistance, type 2 diabetes, or future cardiovascular events [25]. In contrast, brown adipocytes have the ability to maintain normal body temperature in cold environments through heat production in the process of non-shivering thermogenesis [26]. Brown adipocytes, after being activated by cold exposure, start to oxidize their own lipid stores or fatty acids cleared from circulation and other substrates, e.g., glucose, to produce heat and increase the metabolic rate. The unique function of brown adipocytes is due to the expression of UCP1 in the inner mitochondrial membrane [27]. UCP1 has only been found in BAT and is, therefore, an ideal marker for this tissue. Almost all human cell mitochondria (apart from brown fat mitochondria) are responsible for ATP synthesis, involving the use of lipids or glucose. This process is called respiration coupling, where the energy released during the re-oxidation of reduced coenzymes and oxygen consumption is used to phosphorylate ADP into ATP [28]. A part of respiration energy is also lost as heat. Due to presence of UCP1, brown adipocyte mitochondria respire without being forced to phosphorylate ADP; in such a unique circumstance, energy is dissipated as heat [29]. The capacity of BAT to burn fat and enhance energy expenditure could be used as a novel therapeutic tool to combat obesity and metabolic diseases [30].

### 2.1. BAT and Energy Balance

Thermogenesis is generally described as any metabolic process that releases heat, so whole-body thermogenesis is a counterpart of total energy expenditure (EE) [31]. Body temperature can thus be regulated at the level of thermogenesis, e.g., shivering and non-shivering thermogenesis, and heat loss, e.g., sweat production in heat and during exercise [32]. Thermogenesis or total daily energy expenditure (ADMR) can be divided into basal metabolic rate (BMR, roughly 55–65% of ADMR), diet-induced thermogenesis (DIT, about 10% of ADMR), and energy expenditure for physical activity (AEE). BMR remains relatively constant and is mainly determined by lean body mass [33]. A second description of total daily energy expenditure is the use of obligatory and facultative thermogenesis [34]. Obligatory thermogenesis refers to the energy expenditure needed for daily bodily functions, i.e., those needed for cells and organs to maintain their daily living functions. This also includes parts of DIT and AEE that are not needed for extra heat production. Facultative thermogenesis, in contrast, is connected with extra heat production in response to cold and diet—cold-induced thermogenesis (CIT) and DIT, respectively. In terms of human brown fat and its role in energy expenditure, the most essential process is cold-induced thermogenesis. CIT consists of shivering thermogenesis (ST) and non-shivering thermogenesis (NST). In ST, the involuntary contraction of muscles caused by cold exposure is the main contributor to heat production in moderate to extreme cold [35]. ST can increase human energy expenditure by as much as 3–5 times the BMR. Shivering, however, is generally uncomfortable, leads to fatigue, and negatively affects the coordination of our movements. NST is carried out by the activation of brown adipose tissue (BAT) via UCP1 and the skeletal muscle via sarcolipin [36]. The important role of BAT in cold-induced thermogenesis was proven by the inhibition of BAT thermogenesis using nicotinic acid, which resulted in increased muscle shivering to combat cold temperatures [37]. Active BAT can contribute up to 2–5% of the resting metabolic rate in humans [38,39]. Moreover, non-shivering thermogenesis can be maintained without discomfort. Considering this fact, the activation of brown adipose tissue seems to be one of the possible mechanisms responsible for increasing energy expenditure [40], thereby creating a negative energy balance. This may have large health implications, suggesting that the sustained activation of BAT may alleviate obesity and its associated disorders.

The gold standard method for identifying and assessing BAT volume is 18FDG (18 fluoro-deoxy-glucose) PET/CT [41], but BAT glucose metabolism does not accurately reflect BAT thermogenic activity [42]. Notably, intracellular triglycerides TG are the main energy source used initially after cold-induced BAT activation in humans [37,43]. TG is the main fuel used for mitochondrial oxidative metabolism, which is why BAT glucose uptake should be disconnected from thermogenesis [44]. The glucose uptake and metabolism of BAT better illustrates BAT’s insulin sensitivity. A study using another tracer, ^11^C-acetate, which evaluated the Krebs’ cycle rate, more accurately verified the role of BAT in cold-induced thermogenesis in humans [45], showing a two- to three-fold increase in BAT thermogenesis via acclimation to cold [46,47] without a reduction in T2D subjects, despite a major decline in BAT glucose uptake [48]. Studies evaluating BAT oxygen consumption with the use of ^15^O_2_ estimated BAT energy expenditure at a level of 15–25 kcal/day [49]. A second research group reported BAT thermogenesis, with the use of the same tracer, as ~7 kcal/day at room temperature to ~10 kcal/day during mild cold exposure in healthy subjects [50]. These are rather small amounts of data, which could be a result of the current limitations of using 18FDG PET to measure total BAT volume, especially in obese and T2D individuals. The measurement of metabolically active BAT with the use of radiological 3D mapping estimated the highest BAT contribution to thermogenesis at 27–123 kcal per day at room temperature and at 46–211 kcal per day during mild cold exposure [51]. The same study indicated that 4.3% of the total body adipose tissue mass reflects fat tissue, with significant glucose uptake upon cold exposure. The average mass of BAT ranges from 50 to 70 g in adult humans [3]; such an amount of active BAT could increase daily energy expenditure by about 170 kcal. Outcomes from an interventional study in which patients without active BAT at the baseline visit lost weight after six weeks of cold exposure and presented a 1.5-fold increase in BAT activity [52] support the role of brown adipose tissue in energy balance in humans. It was calculated that 63 g of fully activated supraclavicular BAT would utilize an amount of energy equivalent to 4.1 kg of WAT [3]. The appreciable influence of BAT on energy expenditure is supported by an experimental study in which treatment with the β3-adrenergic agonist mirabegron not only increased energy expenditure (203 ± 40 kcal/day), but also increased beneficial lipoproteins (HDL and ApoA1) and antidiabetic proteins (adiponectin), and improved insulin secretion and sensitivity [53].

Recent discoveries have shown the different mechanisms for facultative energy expenditure in BAT, including the futile cycles based on creatine [54] and succinate [55]. In beige adipocytes, cold exposure elicited mitochondrial creatine kinase activity and increased the expression of genes associated with creatine metabolism. Compensatory genes of creatine metabolism are induced when UCP1-dependent thermogenesis is ablated. Succinate accumulated from the extracellular milieu is rapidly taken up by brown adipocyte mitochondria, and its oxidation by succinate dehydrogenase (SDH) is required to activate thermogenesis. It was found that SDH-mediated succinate oxidation initiates reactive oxygen species (ROS) production and thereby drives UCP1-dependent thermogenic respiration, while SDH inhibition suppresses thermogenesis. The findings of these studies were determined using rodent models and still need to be evaluated in humans.

### 2.2. BAT, Obesity, and Insulin Resistance

Negative energy balance is not the only factor in favor of using brown adipose tissue for the treatment of obesity; an additional significant feature is the ability of BAT to alleviate insulin resistance and disturbance in glucose homeostasis. Results from experimental studies showed that active BAT is inversely correlated with the body mass index (BMI), which additionally confirms the relationship between BAT and body mass [56]. Patients with detectable BAT had a lower body mass index and increased energy expenditure. In terms of obesity and BAT, brown adipocytes occur less often in people with central obesity and hepatic fat [5]. Visceral adipose tissue is known to be an active endocrine organ that highly contributes to insulin resistance. The negative association between BAT activity and the amount of VAT is optimistic in terms of the prevention and treatment of metabolic disease through BAT activation [57]. The removal or sympathetic denervation of murine BAT enhances hypertriglyceridemia and obesity [58]. Serum hypertriglyceridemia with subsequent storage in WAT (ectopically stored in skeletal muscle and the liver) reduced the insulin sensitivity of these organs and increased the risk of T2D [59]. Brown adipocyte clear serum from TG is used to refill the lipid stores used for non-shivering thermogenesis. Moreover, in animal models on a high-fat diet, BAT transplantation significantly reduced body weight and adipose tissue inflammation, and increased overall glucose tolerance and insulin sensitivity. The excision of brown adipocytes caused a significant increase in body weight, adipose tissue inflammation, and insulin resistance [60]. Moreover, the observed decline of BAT activity with aging increases excessive fat accumulation [52]. The age-associated decrease in BAT is explained by the loss of mitochondrial functions, UCP-1 expression, impairment of the sympathetic nervous system, and alteration in the function of brown adipogenic stem/progenitor cells [61]. In contrast, recently published findings indicate that the presence of BAT, even in those aged over 60 years old with cardiovascular disease, is still associated with a lower waist circumference and less metabolic dysfunction, such as lower triglycerides, higher HDL-c, and the absence of T2D [62]. Increased mass and activity of BAT after 10 days of cold acclimation (14–15 °C) in eight patients with T2D resulted in enhanced peripheral insulin sensitivity by ~43%, which supports brown adipose tissue as a new approach for diabetes treatment [63].

A less frequent prevalence of BAT was noted in obese subjects compared to lean individuals, indicating the thermogenic effect of brown adipose tissue on body weight reduction [64]. This is also supported by the findings observed in patients after bariatric surgery [65] and in obese subjects after interventional studies [66], in whom weight loss enhanced the glucose uptake of BAT. The above-mentioned data highlight the association between BAT and body weight, with an emphasis on the beneficial effects of BAT in decreasing central adiposity, which is a metabolically harmful status.

Obesity is characterized by the chronic low-grade activation of the innate immune system. In this respect, macrophage-elicited metabolic inflammation and adipocyte–macrophage interactions have a primary importance in obesity [67]. Adipocyte hypertrophy and local hypoxia due to adipocyte expansion are two important contributing factors to the increased accumulation of macrophages in adipose tissue in obesity. These adipocytes promote inflammation via their own cytokine and chemokine synthesis machinery [68]. The secretion of monocyte chemoattractant protein-1 (MCP-1) from adipocytes directly triggers the recruitment of macrophages to adipose tissue [69]. The microenvironment in lean adipose tissue is composed of a 4:1 M2:M1 macrophage ratio. Indeed, diet-induced obesity leads to a shift in the activation state of adipose tissue macrophages from an M2-polarized state, which may protect adipocytes from inflammation, to an M1 proinflammatory state [70]. The primary trigger for the recruitment of M1 macrophages is thought to be the secretion of tumor necrosis factor TNF-α from hypertrophied adipocytes.

Surprisingly, in brown fat depots, the number of macrophages is rather low [16] or even undetectable [71]. Peterson et al. showed that macrophages make up only 30% of the BAT immune cell population, which is already less than 5% of all live cells. A reduction in the BAT macrophage percentage was also reported in obese mice [72]. Notably, macrophage infiltration and the secretion of inflammatory molecules in BAT were found to be significantly lower than those in WAT. Moreover, in the adipose tissue of mice with diet-induced obesity, the activation of classically activated macrophages was shown to suppress the induction of UCP-1 [71].

Brown adipocytes exhibit the intrinsic ability to impair the inflammatory profile of macrophages, while white adipocytes enhance this profile. This suggests that brown adipocytes may be less prone to adipose tissue inflammation, which is associated with obesity [73].

Recently, Fisher et al., demonstrated that the expression levels of almost all macrophage marker genes (M1 and M2) were much lower in the brown fat of mice acclimated to room temperature (middle-aged and young mice) than in the brown fat of thermoneutral mice (30 °C) [74]. Moreover, the expression levels of two genes related to thermogenesis (UCP1 and PGC1alfa) were found to be significantly higher in the brown fat of mice acclimated to room temperature. Neither temperature (thermoneutrality), an energy rich diet, nor increased age were found to be the factor most associated with macrophage accumulation in brown fat. The appearance of macrophages in brown fat coincides with the cessation of its thermogenic activity.

The majority of macrophages found in the brown fat of thermoneutral mice are organized into multinucleate giant crown-like structures. The authors hypothesize that the macrophages found in thermoneutral brown fat perform their conventional (but likely not their only) function: phagocytosis and degradation of dead cells. Brown fat macrophages thus orchestrate tissue remodeling and enable the maintenance of metabolic homeostasis in tissue analogically to the situation in WAT [74].

The brown fat of thermoneutral mice retains full competence during the process of cold acclimation. Thus, profound macrophage accumulation does not influence the thermogenic recruitment competence of brown fat.

### 2.3. BAT, Glucose, and Lipid Metabolism

The activation of brown adipose tissue is triggered by cold exposure. Lower temperatures are detected by skin receptors, which convey signals via the wider neuronal network, including the hypothalamus as a key regulator of body core temperature, through the spinal cord, and finally to the peripheral sympathetic nervous system (SNS) of BAT. Noradrenalin is released after the activation of SNS; it then binds to adrenoreceptors (mainly the B3 receptor) and the lipolysis process of brown adipocytes is initiated. Fatty acids from triglyceride (TG) lipid droplets are the main source of oxidation by UCP1 in brown mitochondria. The reduced amount of TG needs to be restored mainly through the uptake of glucose, albumin-bound free FA, and TG-derived fatty acids from LDL and chylomicrons in the plasma. Findings from animal studies confirm the involvement of BAT in total energy expenditure and TG clearance and metabolism [75]. Moreover, in obese humans, cold exposure resulted in increased fatty acid uptake by BAT compared to muscle and white adipose tissue. BAT volume was significantly associated with lipid metabolism and adipose tissue insulin sensitivity in humans. Functional analysis of BAT and WAT demonstrated the greater thermogenic capacity of BAT compared to WAT, while molecular analysis revealed the cold-induced upregulation of genes involved in lipid metabolism only in BAT [76]. Furthermore, the increase in BAT CT radiodensity observed after acute cold exposure indicates reduced BAT triglycerides, suggesting the use of the BAT internal lipid stores [77]. Consistently, the impairment of intracellular lipids in BAT was found via necropsy in humans who had passed away from hypothermia [78] (Thus, acute BAT activation results in increased fat oxidation from the BAT lipid stores. It was estimated that with a mean total body BAT mass of 168 g [45], a reduction in BAT TG by about 8 g of TG (~72 kcal) was observed over two hours of very mild cold exposure [79], while prolonged BAT activation significantly increased TG clearance from the plasma to replenish intracellular lipids in brown adipocytes. Cold-induced BAT activation with non-esterified fatty acid (NEFA) uptake was shown to be associated with BAT thermogenesis [50] Notably, the disadvantage of using a fatty acid tracer compared to a glucose tracer is that the uptake of the fatty acid tracer is rather nonspecific, since such tracers are also largely taken up by the liver and intestines. Therefore, fatty acid tracers should be improved before they can be used in further research.

The use of the 18F-FDG tracer in imaging studies with PET confirmed glucose uptake by human BAT. The 18F-FDG tracer is a glucose analogue that is transported into cells by the same transporters as glucose. The presence of Glucose transporter type 4 GLUT-4 and GLUT-1 was first identified in brown murine adipocytes, suggesting both the insulin-dependent and insulin-independent uptake of glucose by the tissue [80] In activated BAT, glucose is utilized to refill intracellular lipid droplets and facilitate ATP generation, rather than oxidation, in non-shivering thermogenesis. The glucose clearance capacity of brown adipose tissue after cold exposure was confirmed in several studies; additionally, it was calculated that BAT accounts for ~1% of total body glucose use, compared to ~50% for skeletal muscle [48]. BAT glucose uptake in healthy individuals is responsible for the utilization of 5 g of glucose or ~23 kcal. Brown adipocytes fully oxidize the glucose that is taken up.

The insulin-mediated FDG uptake by BAT suggests that the GLUT-4 transporter participates in glucose uptake in human BAT. Consequently, BAT could be considered as an insulin-sensitive tissue [81,82]. Studies on animal models also confirmed the contribution of BAT to whole-body glucose metabolism [83]. In humans, the activity of BAT is associated with lower glycosylated hemoglobin (HbA1c) [84] and plasma insulin and glucose [85], suggesting that BAT could have an impact on glucose metabolism. In warm conditions, with the use of a hyperinsulinemic euglycemic clamp, insulin was shown to stimulate BAT glucose uptake without stimulating blood flow, suggesting that insulin signaling increases BAT glucose uptake independent of BAT thermogenic activation [80]. However, under cold exposure, as well as under conditions with a high level of insulin, glucose uptake is increased significantly, dissipating energy as a function of increased blood flow [80]. Subjects in whom brown adipocytes have been detected are more insulin sensitive, while in obese patients, despite cold-induced BAT activation, glucose uptake is blunted [66]. All data indicate that BAT activation participates in the regulation of insulin-mediated glucose disposal. Moreover, insulin seems to regulate BAT mass and function via the SNS. In insulin-deprived animal models, the weight and thermogenic capacity of BAT declined, but re-adding insulin restored the function and mass of BAT [86]. In insulin receptor knock-out rodents, a decrease in BAT mass was observed [87]. Additionally, insulin enhances UCP1 expression and thermogenic function in BAT via augmentation of the sympathetic nervous system [88,89]. In diabetic mice, insulin treatment was found to increase the UCP1 expression of BAT. However, denervation of the SNS seems to mediate the UCP1 content in BAT via insulin [90], indicating that an increase in UCP1 and BAT thermogenic function via insulin requires SNS activation.

The decreased activity of BAT, through impaired glucose uptake in resting conditions and reduced glucose clearance from serum, may predispose an individual to T2D. Recently published epidemiological outcomes indicated an association between increased glycosylated hemoglobin and an increased prevalence of diabetes with higher outdoor temperature [91]. Moreover, BAT activation with cold exposure is also associated with improved glucose homeostasis and insulin sensitivity in patients with T2D [64,92].

Another study has shown decreased BAT glucose uptake rates in overweight and T2D subjects vs. young healthy subjects with no reduced uptake of non-esterified fatty acids and thermogenic activity under cold stimulation [48].

Outcomes from a previous publication estimated a mean total volume of 18FDG-positive BAT at 150 mL in healthy adults [44], which could be due to the limitation of BAT metabolism in enhancing systemic glucose clearance. However, recently reported results featuring the three-dimensional mapping of adipose tissue depots with 18FDG PET/CT suggest that the total BAT volume may be much larger (ranging from 510 to 2358 mL) and could significantly influence glucose homeostasis.

### 2.4. BAT Activators

The most common method of activation for brow adipocytes is cold exposure (Figure 1). The outcomes from PET/CT scans performed for various medical reasons have shown that BAT is much more commonly detectable in PET/CT scans during winter than during summer [77,93,94,95]. The 18 FDG uptake of BAT after acute cold exposure before scanning was higher [3,4,56] than that under warm conditions [96,97,98]]. Cold activates brown adipocytes through the sympathetic nervous system (SNS), which was confirmed by an increase in plasma and urinary noradrenaline levels in cold-exposed individuals [99,100]. The denervation of sympathetic nerves suppresses the cold-induced changes in BAT activity in animals [101]. To determine whether cold acclimation would also increase the amount of BAT or increase its efficiency, experimental studies evaluating two to six weeks of cold exposure noted remarkable increases in glucose uptake and the volume of active BAT, but no studies were able to confirm browning in the biopsies of subcutaneous abdominal WAT under these cooling conditions [46,56,102,103].

Cold exposure certainly plays the most significant role in increasing the metabolic rate via BAT activation. However, studies in both animals [104] and humans [105,106] indicated that food intake could also increase whole-body energy expenditure. This process is called diet-induce thermogenesis (DIT). Studies indicated that certain types of macronutrients can alter energy expenditure in different ways, suggesting that protein increases the metabolic rate more significantly than fats and carbohydrates [107]. The intake of food, especially carbohydrates, increases the activity of the sympathetic nervous system [107]. The idea of diet-induced BAT thermogenesis is based on observations showing that increased SNS and BAT metabolic activities in diet-induced obesity are accompanied by lower weight gain than expected from caloric intake in rodents [108]. Recently published results showed no association between BAT activity and volume with quantified ad libitum energy intake or habitual energy intake estimated from 24 h dietary recalls in 102 young healthy humans [109].

Data from experimental studies indicate that some dietary supplements can increase the metabolic rate via the thermic effects caused by BAT; once such supplement is capsaicin derived from chili peppers. This supplement enhanced thermogenesis, accompanied by a reduction in body fat in both animals [110] and humans [111,112]. Capsaicin acts through vanilloid subtype 1 of the transient receptor potential (TRPV1) receptors on adipose tissue, inducing the brite phenotype in pre-adipocytes [113]. It also increases the central sympathetic stimulation of BAT via the activation of gastrointestinal TRPV1 receptors [114]. A similar thermic effect is also exerted by capsinoids, the non-pungent nutrients of capsaicin. Capsinoids acutely alter resting energy expenditure only in BAT-positive subjects (but not in BAT-negative subjects) [115]. Recently, other dietary nutrients, such as polyunsaturated fatty acids (PUFAs), have been determined to induce BAT activation. In animal models, high-fat diets rich in PUFAs affected the expression of UCP1 mRNA in brown adipose tissue [116]. Outcomes from two other studies indicated that supplementation with omega *n*-3 long-chain polyunsaturated fatty acids enhanced thermogenesis via the activation of brown adipose tissue [117,118]. Moreover, a maternal diet rich in polyunsaturated fatty acids was related to larger interscapular brown adipose tissue depots in animals [119,120]. The correct amount of brown and white adipose depots depends on maternal diet during pregnancy, and may be responsible for the development of obesity, insulin resistance, and T2M in children later in life [121].

### 2.5. Brown and Brite Adipocytes 

Classical BAT depots are typically found in animals in interscapular, cervical, peri-aortic, peri-renal, intercostal, and mediastinal areas [122]. The brown adipocytes that are detected via PET/CT in humans, typically in supraclavicular, neck, paravertebral, and suprarenal sites [2], refer to brown-like adipocytes called brite (from brown to white) [123], beige [124], or recruitable [125]] cells rather than classical brown fat (Table 1).

In adults, brown-like adipocytes with typical brown adipocytes are scattered among white adipocytes in supraclavicular regions [51] and are activated under special conditions like exposure to cold, sympathetic agonists, capsaicin, or irisin. The estimated mass of BAT in humans is in the range of 50–70 g [126], with a small amount able to significantly increase energy expenditure. Therefore, scientific interest is focused on how to increase the current amount of BAT and profoundly enhance its activity.

Classical brown adipocytes originate from myogenic factor 5 positive (Myf5^+^) progenitor cells, similar to skeletal myocytes [127]. In contrast, brite adipocytes have been shown to be derived from Myf5 negative (Myf5^−^) progenitor cells, much like white adipocytes [15]. Their morphology (lipid droplet size and mitochondria content) is intermediary between that of classical brown and white adipocytes. Brite adipocytes feature multiple lipid droplets (though often larger than those seen in brown adipocytes), more mitochondria than white adipocytes, and the expression of UCP1 [128,129] (Figure 1). Classical and brite adipocytes differ in their developmental origin, but both seem to contribute to thermogenesis [130]. The methods used to initiate brite/beige cell formation in brown adipocytes are as follows: acclimation to severe/mild cold temperatures [131], β-adrenergic agonist treatment [132], increased PPARγ, diet, and exercise. Stimulating brite/beige adipocyte formation in humans could be another way to improve glucose homeostasis and a strategy to combat obesity.

The molecular factors that significantly participate in the adipogenesis of brown and white adipocytes are PPARγ and CCAAT-enhancer binding protein (C/EBP) [133]. These transcriptional factors are responsible for adipogenic differentiation and the impairment of PPARγ and C/EBP, decreasing BAT recruitment [134]. PPARγ is necessary for adipocyte differentiation and lipid storage and has antidiabetic effects. After RRARγ binds to its receptor, the expression of genes such as C/EBP increases. This heterodimer links to protein-containing PR domain 16 (PRDM16) and stimulates the differentiation of brown fat cells [135]. Subsequently, PPARγ activates its coactivator 1a (PGC-1α), which enhances UCP1 expression [136]. PGC-1α is a coactivator of PPARγ in brown adipocytes; PGC-1α also regulates energy balance and mediates brown fat cell differentiation [137]. The main role of PGC-1α is to enhance the expression of UCP1, respiratory chain proteins, Krebs cycle proteins, and FA oxidative enzymes [138]. The most prominent role in the differentiation of brown adipocytes seems to be played by PRDM16, which directly interferes with PGC-1α, stimulating the interaction between PGC-1α and PPARγ. This protein promotes the switch from white pre-adipocytes to brown adipocytes [139]. Brown pre-adipocytes decrease the expression of PRDM16, resulting in differentiation into skeletal muscle cells [140]. Another essential player in the development of BAT is bone morphogenetic proteins (BMPs). Among the members of the BMP family, BMP7 is characteristic of BAT and enhances the expression of all early regulators of BAT, such as PRDM16, PGC-1α, UCP1, PPARγ, and C/EBP. A BMP7 knockout animal model showed the depletion of UCP1 but the preservation of WAT differentiation. Conversely, the administration of exogenous BMP7 in these mice promoted BAT but not WAT development, with a consistent increase in energy consumption and a lack of weight gain [141]. BMP-7 was also reported to induce the browning of WAT and to improve insulin sensitivity [126]. Other mediators of BAT activation with beneficial effects on glucose tolerance and insulin sensitivity are fibroblast growth factor 21 (FGF-21) and interleukin-6 (IL-6). In animals, following cold exposure, FGF21 expression was reduced in the liver (where it is produced), but enhanced in brown and white adipose tissue, where FGF21 enhances UCP1 expression and the browning of subcutaneous tissue [142]. These findings have been confirmed in humans, where exposure to decreased temperature increases diurnal plasma FGF21 levels, showing a positive correlation with lipolysis and cold-induced thermogenesis [143]. PPARγ transcriptionally controls FGF21, which then acts as an autocrine or paracrine mediator to increase PPARγ transcriptional activity [144]. FGF21, by increasing UCP1 expression, seems to improve glucose metabolism. These findings were supported by the results from an experimental study on obese animals in which the injection of FGF21 reduced adiposity, improved glycemic control, and increased energy expenditure [145]. Interleukin-6 (IL-6), a proinflammatory cytokine mainly associated with insulin resistance and type 2 diabetes, plays an interesting role in BAT activation [146]. In obesity, IL-6 is typically secreted from visceral rather than subcutaneous adipocytes [147]. Notably, in studies evaluating BAT activity under cold conditions, a significant increase in IL-6 secretion was observed [148]. Moreover, the administration of IL-6 was shown to attenuate weight gain and visceral obesity without affecting food intake. Furthermore, in the same study, IL-6 augmented UCP1 expression in BAT via stimulation of the sympathetic nervous system [149]. This was mediated by the phosphorylation of the signal transducer and activator of transcription 3 (pSTAT3). Further confirming the association between IL-6 and brown adipocytes is a study in which BAT transplantation caused increased insulin-stimulated glucose uptake in both brown and white adipocytes in the heart, but not in skeletal muscle. Moreover, this beneficial metabolic profile was decreased when the BAT used for transplantation came from IL-6 knockout mice, suggesting that BAT-derived IL-6 is necessary to achieve the significant effects of BAT transplantation on glucose homeostasis and insulin sensitivity [82]. Interestingly, IL-6 is a cytokine known to be secreted by skeletal muscle in response to exercise, thereby increasing its insulin sensitizing effects [150].

Skeletal muscles during physical activity are the source of another protein named irisin, which has a significant browning effect on white adipocytes in mice. Irisin is myokine-cleaved from fibronectin type III domain containing protein 5 (FNDC5), which is released by the exercising muscle. In animal models, an increase in irisin during exercise or exogenous administration resulted not only in an increase in WAT browning, but also in a significant reduction in body weight and an improvement in glucose tolerance [151]. Outcomes from animal studies suggest that irisin could exert a positive metabolic effect during exercise through the browning of white adipocytes. Several studies have also been performed to evaluate the role of irisin in humans. Some of these studies confirmed an acute increase in plasma irisin level after exercise [152,153,154,155], while others have not [156,157]. Moreover, the results from interventional studies assessing the influence of different types of training on irisin levels are inconsistent. Only one study showed an increase in irisin levels after 10 weeks of endurance training [151], while after 12 weeks of combined endurance and strength training, no positive association between irisin level and physical activity [155] was observed. Moreover, after long-term training with both aerobic endurance and strength endurance regimens, no effect was observed on irisin concentration [153,157,158]. Experimental studies on human adipocytes treated with irisin showed an increased expression of UCP1 mRNA predominantly in classical brown adipocytes rather than white adipocytes [156]. Irisin (encoded by FNDC5) can increase energy expenditure in humans by inducing browning. Moreover, it was noted that FNDC5 gene expression in human muscle biopsies and adipose tissue with circulating irisin levels is correlated with obesity, insulin sensitivity, and T2D. These results showed that circulating irisin could induce browning of human adipose tissue, leading to the improved function and capacity of BAT and enhancing FNDC5 gene expression in adipose tissue. Moreover, a previous study observed decreased circulating irisin concentrations and FNDC5 gene expression in adipose tissue and muscle from obese and T2D subjects, suggesting a loss of brown fat-like characteristics [159]. Therefore, the role of irisin in humans in terms of BAT activation and glucose homeostasis should be further investigated.

## 3. Conclusions and Future Perspectives

The function of activated brown adipose tissue does not only promote negative energy balance, but also alleviates metabolic complications such as insulin resistance, dyslipidemia, and disturbances in glucose homeostasis. There is still debate around how to maintain and activate the significant number of brown adipocytes needed to exert anti-type 2 diabetic effects. Therefore, future studies will involve the search for new BAT activators.

Currently, the best-explored BAT activators are cold exposure and sympathetic nervous system agonists. In terms of the metabolic benefits resulting from BAT activation, the most important future perspective is to investigate the new molecular and environmental factors elucidating BAT activation, as well as the browning of WAT. The re-activation and recruitment of BAT in obese individuals is especially important because such individuals will receive the most benefit. It remains challenging to identify BAT activators that will be effective under conditions other than cold exposure. Moreover, the role of diet and nutrients is worth exploring to find new methods of BAT activation. Thus far, it has been shown that the stimulation of transient receptor potential channels via some food ingredients, such as capsinoids and the catechins found in green tea, exert anti-obesity effects by the activation and acquisition of new brown adipocytes. Another major goal for future research is the identification of brown adipokines or batokines that could be candidates for drug development to treat obesity or metabolic disease.

The influence of BAT on energy expenditure is underestimated due to the small volume of BAT as measured using 18FDG PET. The accurate measurement of BAT volume is a major limitation, especially in individuals with obesity and T2D. The development of novel imaging methods for the precise quantification of BAT volume and activity is required to assess the true potential of targeting BAT thermogenesis to prevent or treat metabolic disorders.

## Figures and Tables

**Figure 1 ijms-22-01530-f001:**
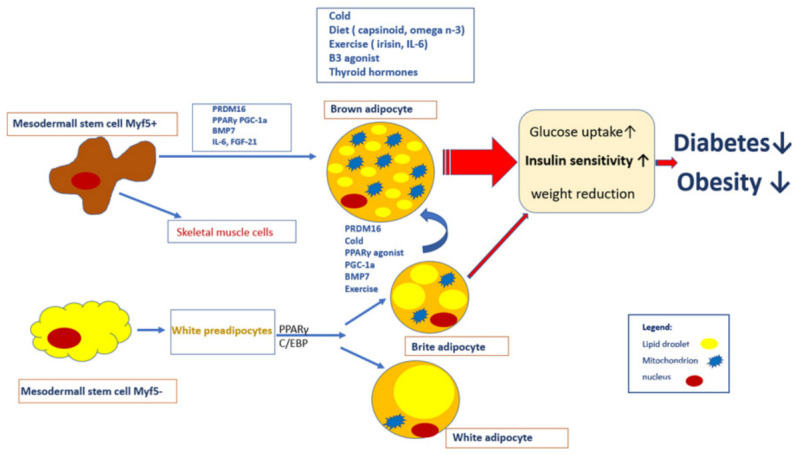
The interplay between brown adipocyte differentiation and activation, as well as insulin and glucose homeostasis. Classical brown adipocytes originate from mesodermal stem cells (Myf5^+^). This process is mediated by the expression of molecular factors such as PRDM16, PPARγ, PGC-1a, BMP7, IL-6, and FGF-21. Moreover, skeletal muscle cells originate from the same precursor cell. Myf5^−^ precursor cells are transformed into white or brite adipocytes. Beige adipocyte differentiation into classical brown adipocytes is induced by molecular factors, e.g., PRDM16 and BMP7, and environmental factors, e.g., chronic cold exposure and exercise, as well as PPARγ agonists. Brown adipocytes are activated by cold, diet, exercise, thyroid hormones, β3 agonists, increased energy expenditure, decreased fat content, and enhanced glucose and insulin homeostasis in humans, leading to a decreased rate of obesity and type 2 diabetes.

**Table 1 ijms-22-01530-t001:** A summary of the main anatomical, cellular, and molecular differences amongst BAT, beige (brite), and WAT and their involvement in obesity and other human disorders.

	Classical Brown Adipocytes	Beige (Brite) Adipocytes	White Adipocytes
Anatomical differences	Mainly in interscapular regions	Scattered among white adipose tissue in the cervical, supraclavicular, and paravertebral regions	Subcutaneous regions, intra-abdominal region, other sites: retro-orbital, bone marrow, pericardial
Cellular differences	Multilocular histology with a central nucleus and an abundant number of large mitochondria, dense vascularity, and innervation by the sympathetic nervous system	Lipid droplet size and mitochondria content is intermediary between classical brown and white adipocytes. Brite adipocytes have multiple lipid droplets and more mitochondria than white adipocytes	Unilocular morphology, triacylglycerols are stored in one large droplet inside the cell
Molecular differences	UCP1-dependent	UCP1-dependent	PPRGγ
	PGC 1α		
	PRDM16	PPRGγ	
Clinical outcomes in humans	Increased metabolic rate, decreased body weight, increased insulin sensitivity	Increased metabolic rate, decreased body weight, increased insulin sensitivity	Energy storage, obesity, type 2 diabetes

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
