# Peer review of "Brown Adipose Tissue and Its Role in Insulin and Glucose Homeostasis"

_ijms, 2021, doi:10.3390/ijms22041530_

Round 1
Reviewer 1 Report
Maliszewska et al., reviewed the role of Brown adipose tissue (BAT) in whole body energy expenditure, lipid and glucose metabolism/homeostasis. The authors also reviewed possible activators of BAT in humans to treat obesity and metabolic disorders.
Overall, the review is nicely written with lots of useful information, but flow needs to be improved. The review in its current form lacks the critical assessment of the resources available for BAT analysis and their usefulness in several disorders. Furthermore, this review needs to be complemented by author discussion and perspective. I have some suggestions to make this review more interesting and appealing to a broader audience.
Major comments:
- Redundant references can be removed, and unnecessary historic details can be cut short.
- Description for Figure 1 is missing in the legend. Without explanation, readers will be guessing what is being presented.
- A table summarizing main anatomical, cellular and molecular differences amongst BAT, Brite and WAT, and their involvement in obesity and other human disorders with references is needed for comparison.
- Many of the comorbidities of obesity, including T2D and cardiovascular diseases, are related to the low-grade chronic inflammation of white adipose tissue. Several reports have suggested that alternatively activated, non-inflammatory, M2 macrophages may play positive roles in BAT activation and WAT browning. Are Toll-like receptors (TLRs) involved in this process? Are TLR agonists/antagonists can be used for BAT activation? Adding a new perspective will make the review impactful.
- A discussion/author perspective to highlight the complexity of the reported findings is required to send a clear message to the reader. Future perspectives section should be added.
- Several lines were redundantly used in the abstract/introduction/conclusion (For ex., lines 12-13 and 45-46). These sentences need to be modified and re-written to express authors viewpoint.
- An overall list of abbreviations will be helpful to the reader. The full form of a lot of abbreviated words is missing throughout the review.
- Rephrase “identify new possible activators”. The authors are not identifying new activators in this review.
Author Response
We wish to thank the Editors and Reviewers for their valuable contributions and comments.
We have submitted a revised version that addresses the comments made by the Reviewers and Editors. Our responses to each comment are detailed below.
I would like to add that the manuscript underwent English language correction by MDPI Editing service.
- Question 1: Redundant references can be removed, and unnecessary historic details can be cut short.
Thank you for your comment. Please see the Introduction section. I’ve removed the redundant references and historic details.
- Question No2 Description for Figure 1 is missing in the legend. Without explanation, readers will be guessing what is being presented.
Thank you for your suggestion. I wrote the description. Please see the description below Figure 1.
Classical brown adipocytes originate from the mesodermal stem cell Myf5 (+). This process is mediated by the expression of molecular factors such as: PRDM16, PPARγ , PGC-1α, BMP7, IL-6, FGF-21. Also skeletal muscle cell originate from the same precursor cell Myf5 (+). The precursor cell deprived expression of Myf (-) are transformed into white or bright adipocytes. Beige adipocyte differentiation into classical brown adipocytes is induced by molecular factors e.g.: PRDM16, BMP7 and environmental, such as chronic cold exposure, exercise, and PPARγ agonists. Activated brown adipocytes, by cold, diet, exercise, thyroid hormones, β3 agonist, increase energy expenditure, decrease fat content and enhance glucose and insulin homeostasis in humans leading to decrease rate of obesity and diabetes type 2.
I’ve corrected the figure 1. I’ve add skeletal muscle cells near mesodermal stem cell Myf5+.
Question No 3
A table summarizing main anatomical, cellular and molecular differences amongst BAT, Brite and WAT, and their involvement in obesity and other human disorders with references is needed for comparison.
Thank you for your comment. Please see a prepared table 1. I’ve inserted it in the section 2.5.
|
|
Classical Brown Adipocytes |
Beige (Brite) Adipocytes |
White Adipocytes |
|
Anatomical differences |
Mainly in interscapular region |
Scattered among white adipose tissue in the cervical, supraclavicular, and paravertebral regions |
Subcutaneous regions, intra-abdominal region, other site: retro-orbital, bone marrow, pericardial |
|
Cellular differences |
Multilocular histological with a central nucleus and an abundant number of large mitochondria, dense vascularity and innervation by sympathetic nerve |
Lipid droplet size and mitochondria content is intermediary between classic brown and white adipocytes Brite have multiple lipid droplets, more mitochondria than a white adipocyte |
Unilocular morphology, triacylglycerols are stored in one large droplet inside the cell |
|
Molecular differences |
UCP1 dependent PGC 1α PRDM16 |
UCP1 dependent PPRGγ |
PPRGγ |
|
Clinical outcomes in humans |
Increase metabolic rate Decrease body weight Increase insulin sensitivity |
Increase metabolic rate Decrease body weight Increase insulin sensitivity |
Energy storage Obesity Diabetes type 2
|
Question No4.
Many of the comorbidities of obesity, including T2D and cardiovascular diseases, are related to the low-grade chronic inflammation of white adipose tissue. Several reports have suggested that alternatively activated, non-inflammatory, M2 macrophages may play positive roles in BAT activation and WAT browning. Are Toll-like receptors (TLRs) involved in this process? Are TLR agonists/antagonists can be used for BAT activation? Adding a new perspective will make the review impactful.
Thank you for your comment. I’ve described the role of macrophage infiltrates in BAT. Please see the section BAT, obesity and insulin resistance (section 2.3).
Obesity is characterized by the chronic low-grade activation of the innate immune system. In this respect, macrophage-elicited metabolic inflammation and adipocyte-macrophage interaction has a primary importance in obesity. Actually adipocyte hypertrophy and local hypoxia due to adipocyte expansion are two important contributing factors to the increased accumulation of macrophages in adipose tissue in the obese state. These adipocytes promote inflammation via their own cytokine and chemokine synthesis machinery. Secretion of monocyte chemoattractant protein-1 (MCP-1) from adipocytes directly triggers the recruitment of macrophages to adipose tissue. The microenvironment in a lean adipose tissue is composed of a 4:1; M2:M1 macrophage ratio. In fact, diet-induced obesity leads to a shift in the activation state of adipose tissue macrophages from an M2-polarized state that may protect adipocytes from inflammation to an M1 proinflammatory state. The primary trigger for the recruitment of M1 macrophages is thought to be the secretion of TNF-alpha from hypertrophied adipocytes.
Surprisingly, in brown fat depots, the number of macrophages is rather low or have not been even detected. Peterson K. et al. showed that macrophages make up only 30% of the BAT immune cell population, which is already less than 5% of all live cells. Moreover, it has also been reported a reduction in BAT macrophage percentage in obese mice. Of interest is that macrophage infiltration and secretion of inflammatory molecules in BAT was found to be significantly lower than in WAT. Moreover, in adipose tissue of diet induce obesity mice, activation of classically activated macrophages has been shown to suppress the induction of UCP-1.
Brown adipocytes exhibit an intrinsic ability to impair inflammatory profile of macrophages, while white adipocytes enhance it. It may suggest that brown adipocytes may be less prone to adipose tissue inflammation that is associated with obesity.
Recently Fisher A. et al. demonstrate that the expression levels of almost all macrophage marker genes (M1 and M2) were much lower in brown fat of mice acclimated to room temperature (middle-aged and standard) than in brown fat of thermoneutral mice (30Ëš C). Also the expression levels of two genes related to thermogenesis (UCP1 and PGC1alfa) were significantly higher in brown fat of mice acclimated to room temperature. Temperature (thermoneutrality), neither an energy rich diet nor increase age, was the factor most associated with macrophage accumulation in brown fat. The appearance of macrophages in brown fat coincides with the cessation of its thermogenic activity.
Majority of macrophages found in brown fat of thermoneutral mice are organized into multinucleate giant crown-like structures.
Authors hypothesize that the macrophages found in thermoneutral brown fat perform their conventional (but probably not their only) function: phagocytosis and degradation of dead cells. Brown fat macrophages thus orchestrate tissue remodeling and enable maintenance of metabolic homeostasis in the tissue analogically to the situation in WAT.
The brown fat of thermoneutral mice retained full competence during the process of cold acclimation. Thus, profound macrophage accumulation does not influence the thermogenic recruitment competence of brown fat.
Question No 5.
A discussion/author perspective to highlight the complexity of the reported findings is required to send a clear message to the reader. Future perspectives section should be added.
Thank you for this suggestion. I added a Future perspectives to the section Conclusion. Please see section 3.
Future perspectives
To date the best explored BAT activators are cold exposure and sympathetic nervous system agonist. In terms of the metabolic benefits resulting from BAT activation, it seems that the most important future perspectives is to investigate for the new molecular and environmental factors elucidating BAT activation but also the browning of WAT. The re-activation and recruitment of BAT in obese individuals, is especially important, because they could benefit most. It still challenging to establish the BAT activators which will be effective under conditions other than cold exposure. Also the role of diet and nutrients is worth to be explored in terms of new ways of BAT activation. So far it has been shown that stimulation of transient receptor potential by some food ingredients such as capsinoids or catechins rich in green tea, have anti-obesity effects via activation and acquiring new brown adipocytes. The another major goal for future research is the identification of the brown adipokines or batokines that are candidates for drug development to treat obesity or metabolic diseases.
The influence of BAT to energy expenditure is underestimated due to its small volume measured using 18FDG PET. The accurate measurement of BAT volume, is a major limitations, especially in individuals with obesity and T2D. The development of novel imaging methods for accurate quantification of BAT volume and activity is required to assess the true potential of targeting BAT thermogenesis to prevent or treat metabolic disorders.
Question No 6
Several lines were redundantly used in the abstract/introduction/conclusion (For ex., lines 12-13 and 45-46). These sentences need to be modified and re-written to express authors viewpoint
Thank you for this suggestion. I’ve re-written it. Please see the section Introduction.
Question No 7.
An overall list of abbreviations will be helpful to the reader. The full form of a lot of abbreviated words is missing throughout the review.
Thank you for this suggestion. I’ve added a list of abbreviations below the main text. Please see lines 640-649.
Question No 8.
Rephrase “identify new possible activators”. The authors are not identifying new activators in this review.
Thank you for this comment. Please see the abstract line 20 and 94. I removed the word “identify” and inserted word “discuss” instead.
Reviewer 2 Report
Manuscript Number: ijms-1059429
Title: Brown adipose tissue and its role in insulin and glucose homeostasis
submitted to: IJMS
In this review, Katarzyna Maliszewska and Adam Kretowski, focuses on the study of BAT in body energy expenditure, lipid and glucose homeostasis and the identification of new possible activators of BAT in humans to treat obesity and metabolic disorders.
This is of interest in the challenge of finding new targets for tackling obesity and associated comorbidities. However, the review carried out requires a more in-depth maturation of the work. A more complete introduction to the topic in the first section, as well as a more detailed description of the molecular mechanisms involved. In general, concepts of interest are discussed but ideas sometimes do not seem to be well linked. It is difficult to follow the thread of the story in some cases. In addition, English need to be improved in depth. Here are some examples that demonstrate what I am talking about.
Major points:
In the first section of the review, would be appreciated a more complete definition of the most relevant molecular mechanisms in BAT, even in comparison with WAT. In many cases the ideas are not well linked. For example, the UCP1 is mentioned without previously defining the importance of this protein in BAT. The last paragraphs of the introduction practically coincide with what has already been commented on the abstract.
Further on, in the section "BAT, obesity and insulin resistance" it would be of added value if the authors described the role of macrophage infiltrates and their relevance also in obesity and inflammatory response. Are there studies that show differences between BAT and WAT tissues in this sense?
Minor points:
- Abstract (line 10) The acronym BAT should be indicated.
- Abstract (line 15) “whole body” it is not a term very apropiated.
- Page 1, line 35: 2009r per 2009
- Some spaces are left over: page 2: lines 56, 63, 64, 70 and so on.
- Excessively repetitive definition of abbreviations. For example: Page 3, line 108
- On page 3, line 151, the "beige adipocytes" are introduced, without having previously defined them. Some phrase indicating their characteristics should be considered.
-On page 4, line 171, the acronym VAT is introduced, which has not been previously defined.
Author Response
We wish to thank the Editors and Reviewers for their valuable contributions and comments.
We have submitted a revised version that addresses the comments made by the Reviewers and Editors. Our responses to each comment are detailed below.
In this review, Katarzyna Maliszewska and Adam Kretowski, focuses on the study of BAT in body energy expenditure, lipid and glucose homeostasis and the identification of new possible activators of BAT in humans to treat obesity and metabolic disorders.
This is of interest in the challenge of finding new targets for tackling obesity and associated comorbidities. However, the review carried out requires a more in-depth maturation of the work. A more complete introduction to the topic in the first section, as well as a more detailed description of the molecular mechanisms involved. In general, concepts of interest are discussed but ideas sometimes do not seem to be well linked. It is difficult to follow the thread of the story in some cases. In addition, English need to be improved in depth. Here are some examples that demonstrate what I am talking about.
Thank you for this suggestions. The paper underwent MDPI language editing proofreading.
Major points:
In the first section of the review, would be appreciated a more complete definition of the most relevant molecular mechanisms in BAT, even in comparison with WAT.
Thank you for this suggestion. Please see the section Introduction. I’ve added:
The protective role of BAT, in terms of metabolic consequences, prompted to molecular exploration of brown adipocytes differentiation. The most relevant molecular factor involved in brown but also in white adipose tissue formation are peroxisome proliferator-activated receptors (PPARs). PPARγ has a crucial role for tissue development and function inducing UCP1 expression during adipogenesis. Moreover, PPARγ agonists may be used to induce browning of white adipose tissue. While PPAR α (PPAR-alpha) activation promotes beige adipogenesis via PGC1α which is a key regulator of mitochondrial biogenesis, adaptive thermogenesis and oxidative metabolism. In the molecular pathways involved in white adipose tissue (WAT) browning, the key factor is PR domain-containing protein 16 (PRDM16) which controls the switch between skeletal myoblasts and brown adipocytes and stimulates adipogenesis by directly binding to PPARγ. Recently, was shown that brown and beige adipocytes released growth and differentiation factor 15 (GDF15) in response to thermogenic activity. GDF15 may mediate downregulation od local inflammatory pathways. Moreover, in the aspect of adipose tissue biology, an important role plays certain microRNAs which regulate BAT and WAT function and differentiation. Such microRNAs regulate white, brown, and beige adipogenesis targeting key transcription factors (e.g., PRDM16, PPARγ, C/EBPB, PGC1α).
For example, the UCP1 is mentioned without previously defining the importance of this protein in BAT. The last paragraphs of the introduction practically coincide with what has already been commented on the abstract.
Thank you for this suggestion. Please see the section Introduction. I’ve corrected it. I’ve added:
BAT is a tissue designed for maintaining body temperature higher than ambient temperatures through heat production primarily via non-shivering thermogenesis. This process is mediated by the expression of the tissue-specific uncoupling protein 1 (UCP1) within inner membrane of the abundant mitochondria. Despite high mitochondria content and high cellular respiration rates, brown adipocytes have a remarkably low capacity for ATP synthesis. Brown adipocytes, in contrast to most human cells, through UCP1 expression and low activity of ATP synthase, diminish the proton gradient by uncoupling cellular respiration, decrease mitochondrial ATP synthesis, to stimulate heat production.
Further on, in the section "BAT, obesity and insulin resistance" it would be of added value if the authors described the role of macrophage infiltrates and their relevance also in obesity and inflammatory response. Are there studies that show differences between BAT and WAT tissues in this sense?
Thank you for your comment. I’ve described the role of macrophage infiltrates in BAT. Please see the section BAT, obesity and insulin resistance (section 2.3).
Minor points:
- Abstract (line 10) The acronym BAT should be indicated.
Thank you for this suggestion. Please see the abstract line 12. I’ve inserted BAT acronym.
- Abstract (line 15) “whole body” it is not a term very apropiated.
Thank you for this suggestion. Please the abstract line 19. I’ve removed whole body and inserted total.
- Page 1, line 35: 2009r per 2009
Thank you for this suggestion. Please see the line 48. I removed the “r”.
- Some spaces are left over: page 2: lines 56, 63, 64, 70 and so on.
Thank you for this suggestion. The paper underwent proof-reading by the MDPI Editing service.
- Excessively repetitive definition of abbreviations. For example: Page 3, line 108
Thank you for this suggestion. I’ve corrected it.
- On page 3, line 151, the "beige adipocytes" are introduced, without having previously defined them. Some phrase indicating their characteristics should be considered.
Thank you for your suggestion. Please see section 2.1 Morphology of brown adipocytes line114.
“Among white adipocytes are scattered beige adipocytes which can be converted to brown-like adipocytes. Browning of WAT is induced by cold exposure or genetic modification. Brite adipocytes share morphological, molecular, and thermogenic characteristics/functions with typical BAT.
-On page 4, line 171, the acronym VAT is introduced, which has not been previously defined.
Thank you for this suggestion. I’ve inserted VAT acronym. Please see the section 2.1 line 110.
Round 2
Reviewer 2 Report
Following the changes made, some concepts that were not clear in the first edition have been clarified. I consider that the work is now good enough for publication. There are no other comments / concerns from my side.